# Systematic Review of Endoscopic Management of Stricture, Fistula and Abscess in Inflammatory Bowel Disease

**Partha Pal \***, **Swathi Kanaganti, Rupa Banerjee, Mohan Ramchandani, Zaheer Nabi, Duvvuru Nageshwar Reddy and Manu Tandan**

Asian Institute of Gastroenterology, Hyderabad 500082, India
* Correspondence: partha0123456789@gmail.com

**Abstract:** Background: Interventional inflammatory bowel disease (IIBD) therapies can play a key role in inflammatory bowel disease (IBD) related stricture/fistula/abscess deferring or avoiding invasive surgery. Methods: A total of 112 studies pertaining to IIBD therapy for strictures/fistula/abscess between 2002 and December 2022 were included by searching Pubmed, Medline and Embase with a focus on technical/clinical success, recurrence, re-intervention and complications. Results: IIBD therapy for strictures include endoscopic balloon dilation (EBD), endoscopic stricturotomy (ES) and self-expanding metal stent (SEMS) placement. EBD is the primary therapy for short strictures while ES and SEMS can be used for refractory strictures. ES has higher long-term efficacy than EBD. SEMS is inferior to EBD although it can be useful in long, refractory strictures. Fistula therapy includes endoscopic incision and drainage (perianal fistula)/endoscopic seton (simple, low fistula) and endoscopic ultrasound-guided drainage (pelvic abscess). Fistulotomy can be done for short, superficial, single tract, bowel-bowel fistula. Endoscopic injection of filling agents (fistula plug/glue/stem cell) is feasible although durability is unknown. Endoscopic closure therapies like over-the-scope clips (OTSC), suturing and SEMS should be avoided for de-novo/bowel to hollow organ fistulas. Conclusion: IIBD therapies have the potential to act as a bridge between medical and surgical therapy for properly selected IBD-related stricture/fistula/abscess although future controlled studies are warranted.

**Keywords:** Crohn's disease; stricturotomy; fistulotomy; endoscopic balloon dilation; self-expanding metal stent

## 1. Introduction

Structural complications of Crohn's disease (CD) like stricture, fistula and abscess occur after initial 4–5 years of disease [1]. At this juncture, in the absence of current effective anti-fibrotic therapy in IBD, interventional IBD and surgery are the mainstays of treatment. These structural complications occur in a specific sequence: chronic inflammation leads to stricture formation which leads to fistula in the upstream bowel along with abscess. Endoscopic stricture therapy depends on basic principles of dilatation (with balloon), cutting (stricturotomy) and stent placement (self-expanding metal stents SEMS) [2]. Endoscopic treatment of fistula initially includes initial treatment of the associated stricture (with aforementioned techniques) and drainage of abscess if any. Then, chronic fistula can be treated with cutting (fistulotomy), filling (with glue/plug/stem cell) or closure (with SEMS/sutures/clips) (Figure 1) [3]. Apart from CD, stricture/fistula/abscess can occur in ulcerative colitis in the post-operative scenario such as after ileal-pouch anal anastomosis (IPAA).

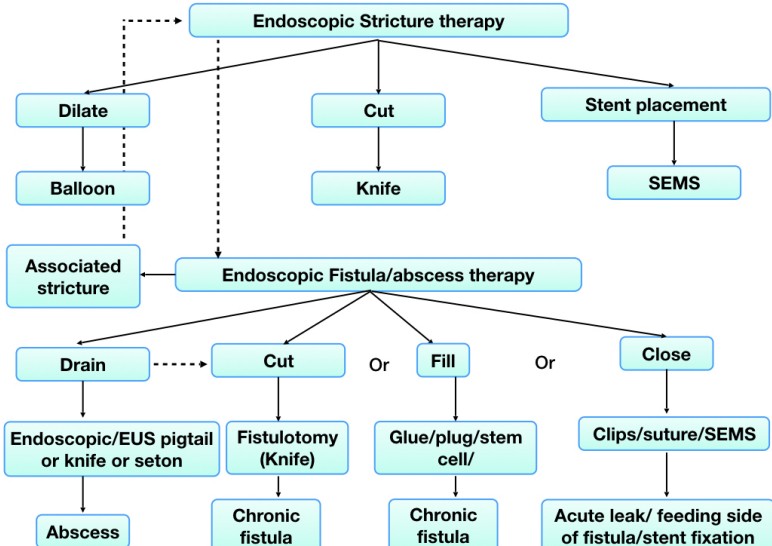

**Figure 1.** Options of endoscopic management for inflammatory bowel disease related stricture, fistula and abscess (SEMS—self expanding metal stent, EUS—endoscopic ultrasound).

## 2. Search Strategy

For the purpose of the review, we searched the PubMed using keywords "inflammatory bowel disease" and "endoscopic stricturotomy" or "endoscopic balloon dilation" or "stent" or "endoscopic fistulotomy" or "glue" or "fistula plug" or "stem cell" or "sclerosing agents" or "endoscopic suturing" or "clips" between 1985 to December 2022. We screened a total of 2927 citations and 259 were identified. Finally, 112 citations were included for our review excluding review articles/consensus guidelines (Figure 2) and including relevant articles with specific searches, those describing novel techniques, and selected cross references. Most of the available literature on endoscopic therapy in IBD is low quality (e.g., retrospective studies, cases series/reports). High quality evidence includes 6 RCTs in this systematic review related to various topics (e.g., balloon dilation versus stenting, stem cell injection for fistula). There were a few uncontrolled prospective studies as well.

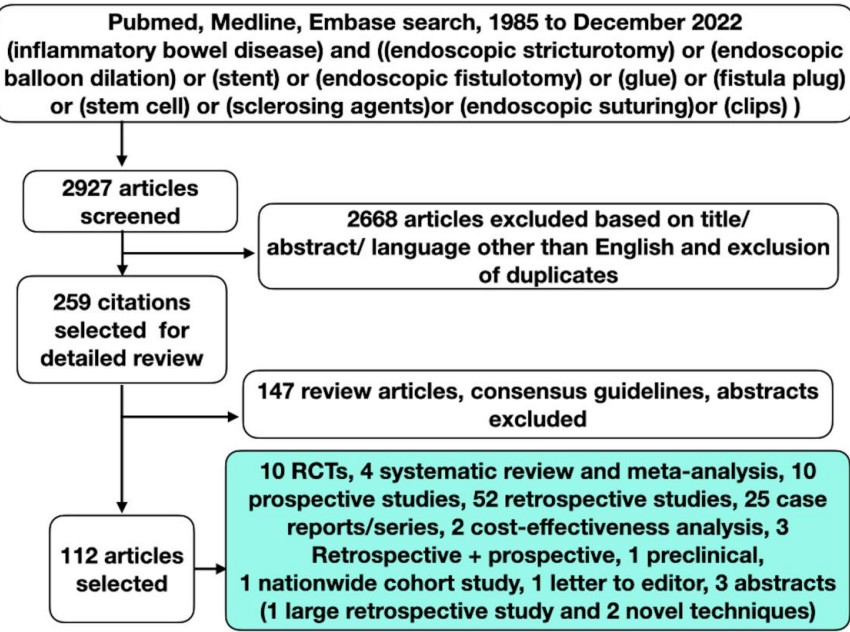

**Figure 2.** Search strategy/PRISMA flow diagram for the systematic review on endoscopic therapy in stricture, fistula and abscess in inflammatory bowel disease (RCT—randomised controlled trial).

### 3. Endoscopic Therapy for Strictures in IBD

*3.1. Endoscopic Balloon Dilation (EBD)*

3.1.1. Outcomes of EBD

EBD in IBD-related strictures is done mostly for CD-related strictures and those related to IBD surgery (e.g., post IPAA in UC and anastomotic stricture in CD) (Table 1). EBD for UC-related strictures should be done after extensive biopsy to rule out malignancy (biopsy can be falsely negative in 3.5%), with a low threshold for surgery due to a high rate of malignancy (0–33%). EBD has high technical success (74–100%), short-term clinical success and low complication rates (0–10.6%) [4–43]. The main drawback of EBD is recurrence. Recurrent symptoms occur in nearly half of the patients [14,44]. Repeated EBD or subsequent surgery is required in nearly two-thirds of patients (21.6–93% and 8–51.7%, respectively, based on existing studies), with follow-up ranging from 20–144 months [44].

Based on a systematic review and meta-analysis, the technical success, short-term clinical success and major complication rates in enteroscopy-guided balloon dilation for small bowel strictures were 94.9%, 82.3% and 5.3%, respectively. Major complications occurred in 5.3% of patients. During follow-up, recurrent symptoms occurred in half (48.3%) and two-thirds required re-intervention (38.8% re-dilation and 27.4% surgery) [44].

3.1.2. Predictors of EBD Success and Surgery-Free Disease Course

In a systematic review, the length of stricture ≤5 cm was shown to be a predictor of surgery-free survival and the risk of surgery increased by 8% with every 1 cm increase in stricture length [27]. Another study showed that stricture length ≥4 cm and inflamed stricture were negatively associated with EBD success [34]. Complications rates were not affected by inflammation [27]. Intra-lesional steroid or anti-tumor necrosis factor (TNF) injections were not associated with the decreased need for re-intervention in another study [17]. Combined anti-TNF and thiopurine therapy was associated with a lower risk of repeat EBD (hazard ratio: 0.23) [24]. On the contrary, another large study has shown that the outcome of EBD is not influenced by concurrent medical therapy or the degree of inflammation [13]. Rutgreet's score of i4 at initial EBD was associated with the risk of anastomotic resection [24]. Anastomotic strictures were associated with better surgery-free survival as compared to de novo strictures [12,16]. One study showed that de novo strictures are more prone to complications. Among various locations of strictures (small bowel, ileo-cecal and gastro-duodenal), gastro-duodenal strictures were associated with a higher risk of recurrent symptoms at 2 years (70.5% compared to less than 50% in other locations) [25,44]. Another study focusing on upper GI strictures showed that repeated dilation was required in 93% [30]. Two studies have evaluated the influence of the diameter of dilation on the risk of subsequent surgery. Dilation diameter ≥15 mm was associated with a successful EBD [34]. A diameter of 14–15 mm had similar surgery-free survival as compared to 16–18 mm dilation, the interval of dilations, however, was longer in the later [29].

Few studies (Table 1) have evaluated the predictive factors for the failure of EBD. One such study designed a nomogram to predict 5-year surgery-free probability after EBD for ileo-colonic anastomotic (ICA) strictures, which included duration of disease, time from surgery, pre-stenotic dilation and symptomatic disease [23] EBD can delay surgery in ICA strictures by more than 6 years. However, the presence of concurrent strictures, history of multiple resections, longer time from the last surgery and shorter time from disease onset were predictors of subsequent surgery [28]. Hence, the patients with aforementioned risk factors can be subjected to upfront surgery rather than EBD deciding on a case-to-case basis. Another such model can predict surgery risk in stricturing ileal CD, known as BACARDI (B3-stricturing disease—1 point, Anti-TNF exposure—1 point, NOD2-CARD15 risk allele- 1 point, pre-stenotic dilation—2 points, inflammatory markers like C reactive protein >11 mg/L—1 point) risk model. A BACARDI risk score of 4–6 predicts the futility of medical/endoscopic therapy and the patient should therefore be subjected to surgery

(Figure 3) [45]. Based on presenting symptoms, presence of obstructive symptoms and absence of perianal involvement were predictive of future surgical intervention [37].

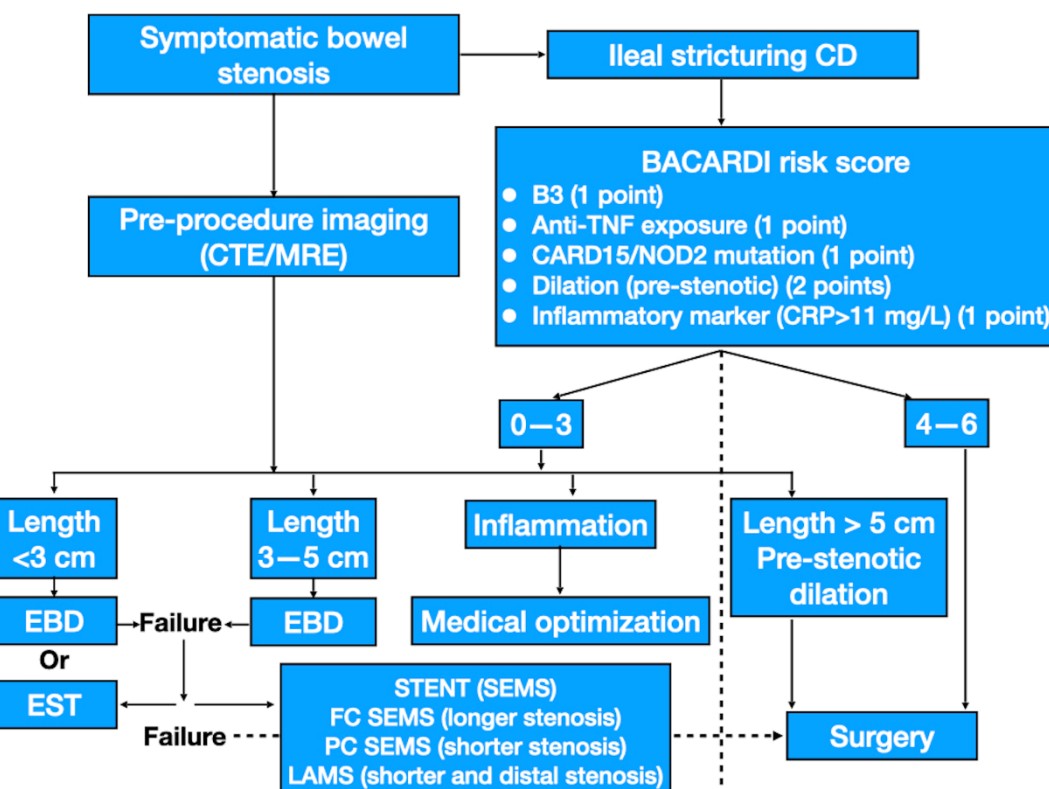

**Figure 3.** Suggested algorithm for the management of Crohn's disease (CD) strictures (CTE—computed tomography enterography, MRE—magnetic resonance enterography, B3—fistulizing CD, TNF—tumor necrosis factor, NOD-2—Nucleotide-binding oligomerization domain-containing protein 2, CARD 15—caspase recruitment domain-containing protein 15, SEMS—self-expanding metal stent, FCSEMS—fully covered SEMS, PC SEMS- partially covered SEMS, LAMS—lumen apposing metal stent, EBD—endoscopic balloon dilation, EST—endoscopic stricturotomy, CRP—C reactive protein).

A meta-analysis included 18 studies (10 full-text articles) (436 patients, 1189 endoscopic balloon dilations) on balloon-assisted enteroscopy-guided dilation of small bowel strictures. The pooled technical success and short-term clinical effectiveness were 94.9% and 82.3%, respectively. However, nearly half (48.3%) of the patients had a recurrence of symptoms, and two-thirds required re-intervention in the form of re-dilation (38.8%) or surgery (27.4%) [44]. The short-term efficacy was lower for de novo strictures (Hazard ratio: 0.40, $p = 0.027$) [44]. Anastomotic strictures can occur due to disease recurrence or surgical techniques whereas de novo strictures are the result of progressive fibrosis, making them more resistant to treatment. Recent studies have looked upon the risk factors of stricture recurrence and subsequent re-intervention which included obstructive symptoms, duration of stricture, stricture length, mucosal healing, location of stricture (small bowel), number of strictures, younger age and use of balloon diameter (<15 mm) [38,39,41–43,46]. Another cost-effectiveness analysis showed the cost-effectiveness of such procedures [47].

More importantly, a recent short-term prospective study and another Danish nationwide study (with over 5 years of follow-up) showed that EBD can prevent surgery in the majority [38,40]. Hence, this has important implications for improving the quality of life.

To summarize, EBD for strictures in IBD has high short-term effectiveness with a high possibility of recurrence on follow-up, which may require re-dilation and surgery. Complications can occur in less than 10% cases.

**Table 1.** Summary of studies on endoscopic balloon dilation in Inflammatory bowel disease/Crohn's disease strictures.

| | Number of Patients | Location, Type of Strictures | Technical Success | Clinical Success | Recurrence Rates | Complications | Repeat Dilation | Surgery on Follow Up | Median Follow Up Period (Months) |
|---|---|---|---|---|---|---|---|---|---|
| Ferlitsch et al., 2006 [4] | 46 | Ileo-colonic Anastomotic | 95% | 89.7% | 62% | 4% | 31% | 33% | 21 |
| Nomura et al., 2006 [5] | 16 | Ileo-colonic, ileo-ileal anastomosis | 93% | 93% | 46.6% | 25% | 25% | 44% | 38.5 |
| Ajlouni et al., 2007 [6] | 37 | De novo and anastomotic Ileo-colonic | 90% | 90% | 32.2% | 3% | 21.6% | 5.4% | 20 |
| Pohl, 2007 [7] | 10 | Small bowel | 80% | 60% | - | 0% | 50% | 40% | 10 |
| Ohmiya, 2009 [8] | 16 | Small bowel | 96% | 100% | n.a | 0% | 12.5% | 18.8% | 16% |
| Despott, 2009 [9] | 11 | Small bowel | 81.8% | 72.7 | n.a | 9.1% | 22.2% | 9.1% | 20.5 |
| Steinecker et al., 2009 [10] | 25 | Lower GI tract (primary or anastomotic) | 97% | 96% | 46% | 3% | 29.2% | 16.7% | 81 |
| Hirai, 2010 [11] | 25 | Small bowel | 72% | 72% | 17% | 8% | 22.2% | 28% | 11.4% |
| Mueller et al., 2010 [12] | 55 | Duodenum, terminal ileum, colon, ileo-colonic anastomosis | 95% | 76% | 9.2% | 1.8% | 47% | 24% | 44 |
| Thienpont et al., 2010 [13] | 138 | Ileal, Ileocolonic | 97% | - | 55.8% | 5.1% per patient analysis | 46% | 24% | 69.6 |
| Gustavsson et al., 2012 [14] | 178 | Anastomotic, upper GI, small bowel, ileo-colonic | 89% | 77% | 66.4% | 5.3% | 66% | 36% | 144 |
| De Angelis et al., 2013 [15] | 26 | Anastomotic, upper GI, small bowel, ileo-colonic | 100% | 81.5% | 54.2% | 0% | 54% | 8% | 40.7 |
| Endo et al., 2013 [16] | 30 | De novo and anastomotic | 93.6% | 93.6% | 60.5% | 10.6% | 60.5% | 37% | 26 |
| Atreja et al., 2014 [17] | 128 | De novo and anastomotic Ileo-colonic | 83% | - | 73.4% | 3.1% | 58.6% | 32.8% | 21.6 |
| Bhalme et al., 2014 [18] | 79 | Anastomotic, upper GI, small bowel, ileo-colonic | 95% | 43% | 66% | 4% | 66% | 23% | 26.8% |
| Chen et al., 2014 [19] | 60 | Anastomosis, ileo-colonic | 94% | - | 16.7% | 0% | 31.7% | 33.3% | 50 |
| Navaneethan et al., 2014 [20] | 8 | Small bowel | 75% | - | - | n.a | 66.6% | n.a | n.a |
| Gill et al., 2014 [21] | 10 | Small bowel | 100% | 80% | - | 20% | 40% | 30% | 16 |
| Hirai, 2014 [22] | 65 | Small bowel | 80% | 80% | - | 4.6% | 50% | 26.2% | 40.3 |
| Lian et al., 2015 [23] | 185 | Ileo-colonic Anastomotic | 91% | - | - | 1.1% | - | 35.7% | 46.8 |
| Ding et al., 2015 [24] | 54 | Anastomotic | 98% | 98% | 68.5% | 1.8% | 68.5% | 18.5% | 72 |

**Table 1.** *Cont.*

| | Number of Patients | Location, Type of Strictures | Technical Success | Clinical Success | Recurrence Rates | Complications | Repeat Dilation | Surgery on Follow Up | Median Follow Up Period (Months) |
|---|---|---|---|---|---|---|---|---|---|
| Guo et al., 2016 [25] | 24 | Upper GI | 92.5% | 95.8% | 79.2% | 8.4% | 79.2% | 24% | 23 |
| Sunada et al., 2016 [26] | 85 | Small bowel | - | - | - | 5.9% | 75.3% | 24.7% | 41.9 |
| Bettenworth et al., 2017 [27] | 1463 | Ileal (98.6%) and anastomotic (62%) | 89.1% | 80.8% | 47.5% | 2.8% | 73.5% | 42.9% | 24 |
| Lian et al., 2017 [28] | 176 | Ileo-colonic Anastomotic | 90.3% | - | - | 8.8%% | - | 51.7% | 21.6 |
| Reutmann et al., 2017 [29] | 135 | De novo and anastomotic Ileo-colonic | 74% | - | - | 0.7% | - | 28.1% | 41.7 |
| Singh et al., 2017 [30] | 35 | Stomach, Duodenum | 93% | 87% | 75% | 4% | 93% | 34% | 15.1 |
| Nishida et al., 2017 [31] | 37 | Small bowel | - | - | - | 8.1% | - | 48.6% | 27.1 |
| Lee et al., 2018 [32] | 30 | Stomach ($n = 1$), small bowel ($n = 5$), colon ($n = 36$) both ulcerative colitis and Crohn's disease | 86.7% | 93.3% | 26.7% | 6.7% | 26.7% | 3.3% | 134.8 |
| Shivashankar et al., 2018 [33] | 273 | Entire GI tract, Pouch, anastomosis | 91.3% | 91.3% | 41.8% | 2.1% | 41.8% | 30% | 31.2 |
| Winder et al., 2019 [34] | 64 | Primary, Anastomotic, Ileo-colonic. | 89.9% | 84.7% | - | 5% | - | 32.8% | 39.6 |
| Chang et al., 2020 [35] | 26 | Ileo-colonic, upper GI | 96.2% | 83.3% | 17.1% | 2.4% | - | 26.9% | 75 |
| Andujar et al., 2020 [36] | 187 | Anastomotic, pouch, ileo-colonic | 79.5% | 55.3% | - | 1.3% | 49.7% | 20.9% | 40 |
| Sivasailam et al., 2021 [37] | 99 | Ileo-colonic, anastomotic | 75% | - | 52% | 3.3% | 52% | 33% | 62 |
| Wewer et al., 2022 [38] | 90 | Small bowel, de novo and anastomotic | - | - | 45.5% | - | 14% | 27% | 60 |
| Watanabe et al., 2022 [39] | 75 | Small bowel, large bowel, anastomosis | - | 78.5% | 68% | 1.1% | - | 40.5% | 82 |
| Pal et al., 2022 [40] | 44 | Upper GI, Small bowel, Large bowel, pouch, anastomosis | 81.8% | 95.4% | 27.3% | 9.1% | 22.7% | 2.3% | 5 |
| Lee et al., 2022 [41] | 114 | Upper GI, Small bowel, Large bowel | 96.4% | 54.3% | - | 0.8% | 16.7% | 18.4% | >6 |
| Ladron et al., 2022 [42] | 32 | Anastomotic | 63.5% | 62.5% | - | 3.2% | 47% | 37.5% | 72 |
| Hibiya et al., 2022 [43] | 98 | Small bowel | 98.3% | - | - | 2% | 75% | 24.5% | 12 |

## 4. Endoscopic Stricturotomy

### 4.1. Method of ES

The second method of endoscopic stricture treatment is endoscopic stricturotomy (ES) by which stricture is cut open by doing a radial incision of stricture with or without the cutting of fibrotic tissue or the placement of spacer clips (i.e., stricturotomy: clips act as spacers to prevent re-approximation and also delay bleeding) [48–55]. ES is done by either a needle knife or an insulated tip (IT) knife. The length of the knives acts as a comparator to decide on the depth of incision (needle knife-5–7 mm, IT knife- 3.5 mm knife and 1.7 mm ceramic tip). The recommended electrocautery settings are Endocut-Q 3-1-3 (effect-3, cut duration-1, cut interval 3). These settings help to minimize the risk of bleeding with ES [2].

### 4.2. Indications of ES Comparison with Other Techniques

ES is usually ideal for short, refractory and fibrotic strictures in the distal bowel, esophagus and stomach. This is because of the fact that the endoscope tip should be under control and the shaft should be straight to control the depth and location of the cutting. Recently, Lan et al. compared ES for primary CD-related distal ileal strictures with ileo-colonic resection. ES had similar surgery-free survival with lower post-operative complication rates (Table 2) [54]. ES is also being increasingly used as alternative primary therapy for short (<3 cm) and anastomotic strictures [48,50]. Novel porcine models of strictures using phenol/trinitrobenzesulfonic acid injection every 2 weeks have been used as training models for this procedure which needs a considerable learning curve [56]. Another large retrospective study compared outcomes of ES ($n$ = 40) with EBD ($n$ − 160) for ileal pouch strictures (inlet or efferent) and showed that both techniques were safe and effective; whereas bleeding and perforation were more common with ES and EBD, respectively [55]. ES requires advanced training and was shown to be having better short-term clinical and long-term efficacy compared to endoscopic balloon dilation with a lower need for re-intervention or surgery (9–22.5%) [48–56]. Perforation rates were reported to be lower (~1%) compared to EBD (1–5%); however, bleeding requiring transfusion can be higher with ES (6–10%, EBD: 3–5%) [48,49,54]. ES and EBD can be combined. A small case series ($n$ = 4) has combined stricturotomy with pulsed argon plasma coagulation with EBD [57].

**Table 2.** Summary of studies on endoscopic stricturotomy (ES) in inflammatory bowel disease (IBD)/Crohn's disease (CD) strictures (EBD: endoscopic balloon dilation).

| Study, Year of Publication | Etiology | Technical Success | Clinical Success | Recurrence Rates | Complications | Repeat Interventions | Surgery on Follow Up | Median Follow Up (Months) |
|---|---|---|---|---|---|---|---|---|
| Lan et al., 2017 [48] | 85 Ileal pouch ($n$ = 50), Crohn's disease ($n$ = 35) (14 combined EBD) | 100% | 54.7% (29/53 with immediate clinical follow up) | 60.6% | 3.7% (bleeding 3.3%, perforation 0.4%) | 60.6% | 15.3% | 11 |
| Lan et al., 2018 [49] | Anastomotic strictures | 100% | 72.7% (vs. EBD 45.4%) | 61.9% | 14.3% (bleeding which required transfusion) | 61.9% | 9.5% | 9 |
| Zhang et al., 2020 [50] | 49 IBD related | 100% | IBD (67.6%) | 34.7% | 4.7% (bleeding) | 49% additional ES, 20.4% additional EBD | 12.2% | 11 |
| Navaneethan U et al., 2020 [51] | 2 Crohn's disease | 100% | 100% | - | 0% | - | - | - |
| Mohy-ud-din et al., 2020 [52] | 11 (IBD, including pouch) | 92% | 92% | - | 9% (self limiting bleeding) | 8% repeat ES | 9% | 5 |
| Moroi et al., 2020 [53] | CD-4 Anastomotic and 1 primary stricture | 100% | 100% | - | 20% (delayed bleeding) | - | - | - |

**Table 2.** *Cont.*

| Study, Year of Publication | Etiology | Technical Success | Clinical Success | Recurrence Rates | Complications | Repeat Interventions | Surgery on Follow Up | Median Follow Up (Months) |
|---|---|---|---|---|---|---|---|---|
| Lan et al., 2020 [54] | Crohn's de novo distal ileal strictures (*n* = 13), (versus ileo-cecal resection, *n* = 32) | 100% | ES (50.0%) (90% with ileo-cecal resection) | 38.5% | 6.9% | 15.4% (surgery) | 15.4% | 21 |
| Lan et al., 2021 [55] | 40 Pouch strictures (vs. EBD-160) | 100% | 42.3% (vs. 13.2% EBD) | 44.4% (vs. 41.3% EBD) | 4.7% (bleeding) (vs. 0.8% EBD) | 22.5% | 22.5% (vs. 20.6%) | 7 |

## 5. Endoscopic Stenting

### 5.1. Indications and Types

Endoscopic stenting for IBD-related strictures is recommended for refractory strictures after the failure of EBD/ES. Although long strictures (3–5 cm) are ideal for endoscopic stenting by the placement of a fully covered self-expanding metal stent (FC SEMS) (Niti S enteral colonic covered stent), shorter strictures can be treated by partially covered SEMS (PC SEMS) (HANARO stent) (see table for various stents and their properties). The latter has lower migration rates compared to FC SEMS. Short, anastomotic strictures can be successfully treated with lumen-apposing metal stents (LAMS). They have short delivery catheters designed for the daring of pancreatic fluid collections and hence are not suitable for proximal stenosis. Bio-degradable stents (SX-ELLA-BD stents) made of polydioxanone degrade in 10–12 weeks and can be used for IBD strictures, although they are still not recommended for routine clinical use. These are not through the scope stents (Table 3) [58,59].

**Table 3.** Summary of various stents used in inflammatory bowel disease (IBD)/ Crohn's disease (CD) strictures.

| Name of Stent | Diameter (mm) | Length (cm) | Stent Type | Specifics |
|---|---|---|---|---|
| Niti S enteral colonic covered stent | 18–22 | 6–15 | Fully covered enteral stent | High migration rates |
| HANARO stent | 20 (26 at ends) | 2.4, 5.4, 7.4 (6, 9, 11) | Partially covered self-expanding metal stent | Lower migration rates |
| Axios stent | 10–20 (21–29 for flanges) | 1 (saddle length) | Lumen apposing metal stent | Short delivery catheter (not for proximal stenosis) |
| SX-ELLA-Biodegradable stents | 18, 20, 23, 25 (23, 25, 28, 31) | 6, 8, 10 | Biodegradable stent | Not through the scope (TTS), made of polydioxanone, degraded in 10–12 weeks |

### 5.2. Technical Tips for Endoscopic Stenting in IBD

The selection of the stent is the first step; it should be at least 1.5 cm longer than the stricture on each side as the stents can undergo a shortening of up to 5–40%. Stricture length should be assessed by injecting radiographic contrast material through a catheter/Fogarty balloon after passing a hydrophilic soft guidewire through the stricture. After stent placement, it needs to be fixed by thought-the-scope (TTS) clips, over-the-scope clips (OTSC) or endoscopic suturing. The duration of stenting should not be longer than 4 weeks for FCSEMS and 1 week for PCSEMS.

*5.3. Results of Endoscopic Stenting*

According to a recent meta-analysis, the pooled rates of technical and clinical success of endoscopic stenting were 93% and 61%, respectively. Main drawbacks were stent migration (pooled rate: 43.9%, 6.4% proximal), pain abdomen (17.9%) and perforation (2.7%). Repeated stenting was required in 9% (Table 4) [60–84]. Another recently published meta-analysis of 10 studies has shown similar results except for the fact that PCSEMS was associated with lower stent migration and stricture recurrence rates [85]. In a recent randomized controlled trial (RCT) (ProtDilat), endoscopic stenting was shown to be inferior to EBD with respect to the need for re-intervention after a year (FCSMS-49%, EBD-20%, odd ratio-3.9). Another RCT was terminated early due to increase in adverse events in the stent arm although clinical success was higher compared to EBD [82]. Inspite of the negative results, endoscopic stenting may have a role in refractory and long strictures related to IBD/Crohn's disease (CD) [83].

**Table 4.** Summary of studies on endoscopic stent placement in inflammatory bowel disease/Crohn's disease (CD) strictures.

| Author/Year | No. of Patients | Length | Stent Type | Technical Success | Clinical Success | Recurrence | Adverse Events/Migration | Repeat Intervention | Duration of Stenting (Weeks) | Surgery | Follow Up (Months) |
|---|---|---|---|---|---|---|---|---|---|---|---|
| Whole et al., 1998 [60] | 1 Colon, CD | - | Tracheo-bronchial Wallstents | 100% | 100% | - | - | 100% | 3 | Used as bridge to surgery | 0.75 |
| Matsuhashi et al., 2000 [61] | 2 Colon, IC Anastomosis, Post EBD | - | FCSEMS (specially modified) | 100% | 100% | 0% | 100% (migration) | 0% | 4 and 22 | 0% | 54 |
| Suzuki et al., 2004 [62] | 2 Colon | - | USCEMS | YES | yes | Yes | Fistula in 1 | Surgery and repeat stenting | 3 and 104 | 1/2 | 3 and 26 |
| Bickston et al., 2005 [63] | 1 ileo-cecal Post EBD | - | 2 UCSEMS | yes | yes | - | - | - | 8 | Used s bridge to surgery | 2 |
| Wada et al., 2005 [64] | 1 Colon | - | UCSEMS | Yes | yes | Restenosis | Perforation, fistula | Yes- surgery | 139 | Yes | 8 |
| Dafnis et al., 2007 [65] | 1 colon | 5 cm | 4UCSEMS | yes | yes | Yes | - | 4 times | 14 | No | 1 |
| Martines et al., 2008 [66] | 1 IC anastomosis Post EBD | 6 cm | FCSEMS | Yes | yes | - | - | - | 1 | Used as bridge to surgery | 0.25 |
| Small et al., 2008 [67] | 1 rectum | - | 2 PCSEMS | yes | yes | - | - | - | 1 | Used as bridge to surgery | - |
| Keranen et al., 2010 [68] | 2 Anastomosis | - | FCSEMS UCSEMS | yes | yes | - | Perforation-1 | Surgery 1 | 6 and 221 | 1/2 | - |
| Rejchrt et al., 2011 [69] | 11 CD Post EBD 07 | 1.5–5 | Polydioxanone biodegradable stent | 90% | 63% | 36.3% | 27% early stent migration | - | 16 | - | 16 |
| Attar et al., 2012 [70] | 11 CD Post EBD-9 | 1–4 cm | FCSEMS | 90% | 36% | 63.6% (1 year), total 90% | 10% proximal migration, 70% migration | 18.2% | <4 | 18.2% | 26 |
| Branche et al., 2012 [71] | 7 CD Ileo-colonic (IC) anastomosis Post EBD | <5 cm | PCSEMS | 100% | 71.4% | 28.5 | 42.8% pain | 14% (EBD) | 1 | 0% | 10 |
| Levin et al., 2012 [72] | 5 IC anastomosis Post EBD-2 | <5 cms | UCSEMS | 100% | 80% | 20% | 0% | 20% | 3 (1 patient at 9 years) | 20% | 28 |
| Loras et al., 2012 [73] | 17 CD Post EBD 14 | 2–6 cm | PCSEMS/FCSEM | 94.1% | 64.7% | 31% | 5.9% spontaneous migration 52% migration | - | Mean-4 | 43.7% | 12 |
| Karstensen et al., 2016 [74] | 6 CD Post EBD | 2–10 | Polydioxanone monofilament, biodegradable stent | 83% | 20% | 80% | 17% stent migration | - | - | - | 4–42 |

**Table 4.** *Cont.*

| Author/Year | No. of Patients | Length | Stent Type | Technical Success | Clinical Success | Recurrence | Adverse Events/Migration | Repeat Intervention | Duration of Stenting (Weeks) | Surgery | Follow Up (Months) |
|---|---|---|---|---|---|---|---|---|---|---|---|
| Axelrad et al., 2018 [75] | 1 Rectal-colon anastomosis Post EBD | 1 cm | LAMS | Yes | Yes | 0% | No | - | 8 | - | 3 |
| Oztas et al., 2018 [76] | 1 IC anastomosis | 3 cm | UCSEMS | Yes | Yes | Yes | 0% | Yes, (PC-SEMS within FC SEMS) | 24 (UC SEMS), 52 (PC SEMS) | - | 12 months |
| Ouali et al., 2019 [77] | 1 Pouch inlet stricture Post EBD/ES | 10 cm | FC-SEMS | yes | yes | Yes | spontaneous migration | EBD, ES | 1 | 0% | - |
| Fung et al., 2020 [78] | 1 Descending colon | - | UCSEMS | yes | yes | No | spontaneous migration | No | < 1 | No | 10 |
| Das et al., 2020 [79] | 21 CD | <6 cm | PCSEMS | 95.8% | 54.2% | 12.5% | 21.7% (2 pain, 3 migration) | 9.5% restenting | 1 | - | 3–50 |
| Lamazza et al., 2021 [80] | 4 rectum Post EBD | - | FCSEMS | 100% | 100% | 75% | 25% migration | 75% (2 EBD, 1 surgery) | 2–12 | 25% | 12 |
| Attar et al., 2021 [81] | 46 CD Post EBD-36 | Mean 3.9 cm (all <5 cm) | PCSEMS (Hanaro stent) | 100% | 58.7% | 6.5% | 15.2% (4 pain, 3 proximal migration) | 34.8% | 1 | 17.3% | 26 |
| Heden strom et al., 2021 [82] | 7 CD | - | PC SEMS | 100% | 86% | - | 71.4% (4 pain, 1 bleeding) | 14.2% | 1 | 14.2% | 69 |
| Andújar et al., 2022 [83] | 39 CD | Mean 4 cm (all <9 cm) | FCSEMS | 92.3% | 51% | - | 7.7% (2 proximal migration, 1 perforation) | 49% | <1 | - | 12 |

EBD—endoscopic balloon dilation, FCSEMS—fully covered self-expanding metal stents, UCSEMS—uncovered SEMS, PCSEMS—partially covered SEMS, ES—endoscopic stricturotomy, IC—ileo-colonic.

## 6. Endoscopic Management of Fistula and Abscesses

Fistula in CD can be associated with stricture and abscess, as they occur in the following sequence: stricture > fistula > abscess. Associated stricture and abscess can be treated endoscopically with EBD/ES/stenting and endoscopic incision and drainage by needle knife/seton placement/endoscopy or endoscopic ultrasound (EUS) guided drainage of pelvic abscess (this can be useful if the radiological approach is difficult due to overlying bowel loops) [3]. CD-related de novo fistula and those from gut to hollow organs (bladder, vagina) should be treated with surgery, whereas short (<3 cm), superficial, simple, bowel-to-bowel (distal), and pouch-to-pouch fistulas can be treated endoscopically. The endoscopic treatment modalities in the order of preference are: cutting (fistulotomy), injection of filling materials (glue, fistula plug, stem cells, sclerosing agents) or closure (using clips/stents/suture) [3].

### 6.1. Endoscopic Drainage

Endoscopic drainage for abscesses associated with CD-related fistula can be done with a needle knife for perianal fistula and intra-abdominal and pelvic abscesses. For perianal fistula, complete fistulotomy can be done for a short, superficial fistula outside the external anal sphincter using a needle knife whereas partial fistulotomy is reserved for the long fistula [3]. Intra-abdominal/pelvic abscess is not feasible for drainage by interventional radiology due to overlying bowel but can be drained with endoscopic pigtail drainage (with or without EUS guidance) [3].

### 6.2. Endoscopic Fistulotomy

Endoscopic fistulotomy in IBD can be done for postoperative bowel-bowel fistula (fistula at suture line, anastomotic leak into distal bowel), pouch-pouch fistula, perianal fistula and primary ileo-cecal fistula. For endoscopic therapy, short (<2 cm), superficial, distal bowel fistulas are ideal. The largest case series to date (*n* = 29) have described the feasibility of fistulotomy mainly in perianal (n-6), tip of J fistula to anastomotic site fistula (*n* = 7), pouch-to-pouch fistula (*n* = 14) and others (neo-terminal ileum to pouch body, fistula from ileo-colonic anastomotic site to colon) [86]. Fistula resolution and clinical success were reported in 89.6% and 75.8%, respectively. A patient had post-procedural bleeding and none had a perforation [86]. Other case reports also described fistulotomy for pouch-pouch fistula and enteroentero-cutaneous fistula [87–89]. After fistulotomy, endoclips can be placed to prevent re-approximation of the fistula tract. Fistulotomy is an option of short, superficial, simple, bowel-bowel/pouch-pouch fistula, whereas it should be avoided in long, deep fistulas, those located close to sphincters or anterior rectal nerves (due to proximity to genital structures).

### 6.3. Injection of Filling Materials

6.3.1. Glue

Fibrin glue was first evaluated as an adjunctive treatment with an anal advancement flap for complex anal fistula repair and was shown to be no more effective than an anal advancement flap alone in a randomized controlled trial (RCT) [90]. Another RCT compared glue injection with observation in low anal fistulas after seton removal. Fistula healing was better in the glue arm (38%) compared to the observation arm (16%). This benefit was most pronounced in simple fistulas [91]. Another retrospective study in 119 patients showed that fibrin glue injection led to complete fistula remission in 45.4% at 1 year (63% in those on combined immunomodulators and biologic therapy) [92]. A randomized controlled trial comparing seton removal alone with seton removal and glue injection showed that seton removal alone was not inferior to the closure with glue [93].

6.3.2. Fistula Plug

A prospective study in 20 patients (36 fistula tracts) has shown that an anal fistula plug (AFP) was successful in closing Crohn's anorectal fistula in 80% of patients and 83%

of fistula tracts (higher with simple fistula) [94]. In contrast, another RCT did not show any benefit of AFP over seton removal alone for CD anorectal fistula [95]. A long-term follow-up study (median follow-up of 110 months) showed an overall healing rate of 38%. No additional benefit was seen after the use of three fistula plugs [96]. However, AFP is usually placed in operation theatre by surgeons; although it can be done under endoscopic guidance.

### 6.3.3. Stem Cells

Adipose tissue-derived allogenic stem cell injection (120 million cells) into the fistula has shown to be effective in inducing clinical and radiologic remission at 24 weeks (51% versus 36% placebo) followed by maintaining remission at 52 weeks (56.3% versus 38.6% placebo) in CD-related complex perianal fistula refractory to conventional and biologic therapy (ADMIRE CD trial) [97,98]. Long-term results of the study (INSPECT study) at two and three years showed a sustained response of 65.5% and 54.2%, respectively [99,100]. Although done by surgeons in the aforementioned studies, endoscopic stem cell injection is feasible as shown in recent studies [101,102]. This strategy has been shown to be more cost-effective than a fecal diversion in refractory fistulas [103]. In single-tract perianal fistulas, stem-cell loaded fistula plugs can be used [104,105].

### 6.3.4. Sclerosing Agents

An amount of 10 mL each of 50% dextrose and doxycycline injection into the chronic non-healing sinus in the rectal stump post J pouch surgery for three sessions has been shown to induce fibrosis and facilitate healing [106].

### *6.4. Endoscopic Closure*
### 6.4.1. Endoscopic Clipping

Over-the-scope clips (OTSC) (designed for gastrointestinal defect closure) rather than through-the-scope (TTS) clips (designed for bleeding control) are more effective for IBD surgery-related anastomotic leaks. OTSC is more useful for leaks/perforations rather than fistula. Case reports and series have described the use of OTSC for the successful treatment of leaks at the tip of J and perianal fistulas (nearly 70% overall technical success), respectively [107,108]. However, OTSC is not recommended for CD-related primary/de novo fistula and bowel-to-hollow organ fistula (rectovaginal and pouch vaginal—TTS can be used for temporary closure) due to suboptimal success and risk of fistula worsening (due to thin septum between pouch/rectum and vagina), respectively. OTSC for enterocutaneous fistula (ECF) can be used on the feeding side of the intestine, with cutting at the exiting side of the skin for adequate drainage. Various case reports/series have described the feasibility and efficacy of OTSC for recto-vaginal fistula (RVF), ano-vaginal fistula, enterocutaneous fistulas and ileal pouch or staple line leaks [109–113]. Moreover, a case report has described the management of refractory rectal fistula with endoscopic submucosal dissection and OTSC [114]. At 16-month follow-up after OTSC for RVF, one-fourth required intestinal resection and 37.5% of patients maintained fistula closure in a small case series [113]. However, the overall results are not very encouraging. Hence, OTSC is recommended only for the closure of surgery-related leaks/perforations with a single tract and minimal/no inflammation [3].

### 6.4.2. Endoscopic Suturing

Endoscopic suturing as a closure method has been described for non-IBD fistulas. There are no reported series for IBD-related fistulas although it is not recommended for bowel-to-hollow organ fistulas (recto-vaginal and poucho-vaginal) and proximal bowel fistulas (technically difficult to reach) [3]. Suturing can be used for IBD endoscopic procedures related to large perforation closure or SEMS fixation.

### 6.4.3. Endoscopic Stenting

FCSEMS have been used for post-surgical strictures and fistula related to CD (3 cases out of a total of 20 case series) [115]. However, stent migration is a major drawback, and long-term efficacy is unknown.

### 7. Conclusions

Endoscopic therapy for strictures, fistulas and abscesses in IBD (Figure 4) is challenging due to the fact that the bowel is often diseased/inflamed/fibrotic, the transmural nature of the disease with extensive submucosal fibrosis, altered bowel anatomy, poor bowel preparation, poor nutritional status and concurrent biologics/steroid use. However, IIBD has the potential to delay or prevent surgery and help manage post-operative complications. The limitations of this systematic review include a qualitative review with a paucity of randomized controlled trials. Most of the evidence related to endoscopic therapy in IBD-related complications is limited to case series/reports and retrospective studies with few controlled studies. Future prospective controlled studies with a comparison with the standard of care can help decide the proper positioning of these approaches in the current treatment algorithm of IBD.

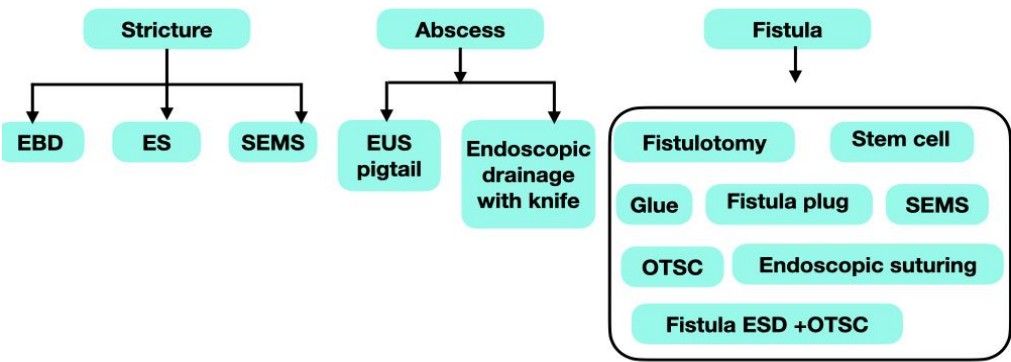

**Figure 4.** Summary of endoscopic therapies for stricture, fistula and abscess in inflammatory bowel diseases (SEMS—self-expanding metal stent, EUS—endoscopic ultrasound, ESD—endoscopic submucosal dissection, OTSC—over the scope clips, ES—endoscopic stricturotomy, EBD—endoscopic balloon dilation).

**Author Contributions:** Conceptualization: P.P.; literature review and writing original draft: P.P. and S.K.; illustrations: P.P.; proofreading and critical review: M.T., R.B., Z.N., M.R. and D.N.R.; approving final manuscript: P.P., S.K., R.B., M.R., Z.N., D.N.R. and M.T. All authors have read and agreed to the published version of the manuscript.

**Funding:** This research received no external funding.

**Institutional Review Board Statement:** Institutional review board approval exempted as the article is a systematic review of previously published literature.

**Informed Consent Statement:** Not applicable.

**Data Availability Statement:** Not applicable.

**Conflicts of Interest:** The authors declare no conflict of interest.

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
