# Peer review of "Systematic Review of Endoscopic Management of Stricture, Fistula and Abscess in Inflammatory Bowel Disease"

_gastroent, doi:10.3390/gastroent14010006_

Round 1

Reviewer 1 Report

The manuscript entitled "Systematic review of endoscopic management of stricture, fistula and abscess in Inflammatory bowel disease" by Partha Pal and co-authors tries to give a systematic overview about endoscopic treatment options in complicated IBD. Indeed, the authors address an interesting and important issue in the daily treatment of patients suffering from IBD, and the evidence level has to be improved. 

However, there are some concerns before the manuscript can be recommended for publication:

- The authors only used Pubmed for literature research. It would be helpful to use another database such as Emboss oder Medline.

- IBD has multiple faces of disease activity and location. Therefore it would be helpful to consider these facts if results are reported about the different techniques. For example,  in my opinion it has to be distinguish between primary stenosis due to stricturing disease character and anastomotic stenosis due to disease recurrence.

- The authors should add a paragraph describing quality of included studies and limitations of data source (e.g. case reports, ...).

- The authors concluded that endoscopic techniques are helpful to delay surgical therapy. This should be discuss in more detail since the primary goal of each therapy should be patients outcome and quality of life. 

- A final figure summarizing therapeutic endoscopic option would be desirable. 

Author Response

The manuscript entitled "Systematic review of endoscopic management of stricture,

fistula and abscess in Inflammatory bowel disease" by Partha Pal and co-authors

tries to give a systematic overview about endoscopic treatment options in

complicated IBD. Indeed, the authors address an interesting and important issue in

the daily treatment of patients suffering from IBD, and the evidence level has to be

improved.

However, there are some concerns before the manuscript can be recommended for

publication:

The authors only used Pubmed for literature research. It would be helpful to use

another database such as Emboss oder Medline.

Thanks for the suggestion. We have now included Emboss and Medline search as well as made the systematic review upto date screening 123 new references and adding 23 new citations. However we did not include conference abstracts in the systematic review and included only full text original articles.

IBD has multiple faces of disease activity and location. Therefore it would be

helpful to consider these facts if results are reported about the different

techniques. For example, in my opinion it has to be distinguish between primary

stenosis due to stricturing disease character and anastomotic stenosis due to

disease recurrence.

Thank you for highlighting this. We acknowledge the fact and added the results

on difference in short term, efficacy of EBD which is higher for anastomotic

strictures than de novo strictures. Also we included other factors like mucosal

healing (disease activity) or location (e.g. small bowel) in predicting outcomes of

endoscopic therapy in IBD.

The authors should add a paragraph describing quality of included studies and

limitations of data source (e.g. case reports, ...).

We have commented upon this. Earlier we had only shown this under PRISMA

diagram only. We elaborated the same in the text as well.

The authors concluded that endoscopic techniques are helpful to delay surgical

therapy. This should be discuss in more detail since the primary goal of each

therapy should be patients outcome and quality of life.

We have included results of long term Danish nationwide study which showed

that surgery can be prevented in majority with endoscopic therapy over 5 years

follow up. We have highlighted this important fact.

A final figure summarizing therapeutic endoscopic option would be desirable.

Thank you for the suggestion. We have added figure 4 summarising all the

endoscopic therapies for stricture fistula and abscess in IBD.

Reviewer 2 Report

I commend the authors for this excellent review and discussion of endoscopic management of stricture,  fistulae, and abscess in Inflammatory bowel disease. This is a well-researched and written paper and provides actionable clinical value to the reader.

Author Response

Thank you so much for your comment. We hope that the review is helpful to the

readers.

Author Response

Thank you for your comments. We hope that the review is helpful to the readers. The variable follow up of patients included in various studies have been included in the tables.